# Quantitative 3D determination of self-assembled structures on nanoparticles using small angle neutron scattering

Zhi Luo[1], Domenico Marson [2], Quy K. Ong[1], Anna Loiudice[3], Joachim Kohlbrecher[4], Aurel Radulescu[5], Anwen Krause-Heuer[6], Tamim Darwish[6], Sandor Balog[7], Raffaella Buonsanti[3], Dmitri I. Svergun[8], Paola Posocco[2] & Francesco Stellacci [1]

The ligand shell (LS) determines a number of nanoparticles' properties. Nanoparticles' cores can be accurately characterized; yet the structure of the LS, when composed of mixture of molecules, can be described only qualitatively (e.g., patchy, Janus, and random). Here we show that quantitative description of the LS' morphology of monodisperse nanoparticles can be obtained using small-angle neutron scattering (SANS), measured at multiple contrasts, achieved by either ligand or solvent deuteration. Three-dimensional models of the nanoparticles' core and LS are generated using an ab initio reconstruction method. Characteristic length scales extracted from the models are compared with simulations. We also characterize the evolution of the LS upon thermal annealing, and investigate the LS morphology of mixed-ligand copper and silver nanoparticles as well as gold nanoparticles coated with ternary mixtures. Our results suggest that SANS combined with multiphase modeling is a versatile approach for the characterization of nanoparticles' LS.

[1] Institute of Materials, École Polytechnique Fédérale de Lausanne, 1015 Lausanne, Switzerland. [2] Department of Engineering and Architecture and INSTM Trieste Unit, University of Trieste, 34127 Trieste, Italy. [3] Institute of Chemical Sciences and Engineering, École Polytechnique Fédérale de Lausanne, 1015 Lausanne, Switzerland. [4] Laboratory for Neutron Scattering and Imaging, Paul-Scherrer Institute, 5232 Villigen, Switzerland. [5] Jülich Center for Neutron Science, JCNS at Heinz Maier-Leibnitz Zentrum, Forschungszentrum Jülich GmbH, 85747 Garching, Germany. [6] The National Deuteration Facility, Australian Nuclear Science and Technology Organisation, Kirrawee DC, NSW 2232, Australia. [7] Adolphe Merkle Institute, University of Fribourg, 1700 Fribourg, Switzerland. [8] European Molecular Biology Laboratory, Hamburg Unit, EMBL c/o DESY, 22603 Hamburg, Germany. Correspondence and requests for materials should be addressed to F.S. (email: francesco.stellacci@epfl.ch)

Self-assembled monolayer-protected nanoparticles have been employed in numerous research fields like catalysis, electronics, and biology[1,2]. One of the keys to their success is the ligand shell (LS) that provides nanoparticles with various functionalities. When multiple ligands are used, nano-domains with complex morphologies, such as patchy and Janus, would appear[3]. Such LS morphology plays an important role in a number of nanoparticles' properties, e.g. interfacial energy[4], molecular recognition[5], and self-assembly[6].

Great efforts have been devoted to understand the thermodynamic phenomena of the LS organization on nanoparticles[7]. For example, Glotzer and co-workers revealed the origin of phase separation of mixed ligands through atomistic and mesoscale simulations[8,9]. It was found that a balance between the enthalpy of phase separation and the conformational entropy leads to the formation of stripe-like domains. The entropy component could arise either from the length mismatch or from the bulkiness of the ligands. Many more calculations were performed afterwards to elaborate such findings and the molecular models are becoming more and more realistic[10–12].

Compared to computational studies, experimental characterizations lack far behind. Scanning tunneling microscopy (STM) has been the main technique to observe LS morphology on nanoparticles[13–15]. However, as common to microscopy techniques, only nanoparticles with clear LS structures can be imaged by STM. In the case of LS with stripe-like morphology, the resolution achieved by STM has been molecular[13] or very close to it[14]. Yet, such resolution was obtained on a few nanoparticles, and consequently the standard deviation for the length scales extracted was large (no better than 30%). Furthermore, when complex morphologies are present (e.g., micellar), the imaging of a limited part of the top of nanoparticle surfaces is not sufficient to extract all of the needed structural parameters to determine quantitatively the morphology in question. There are other alternative techniques capable of characterizing the LS (such as nuclear magnetic resonance (NMR), Fourier transform-infrared spectroscopy (FTIR), and matrix-assisted laser desorption/ionization (MALDI)-time of flight) but they are only qualitative[16], i.e., the LS morphology can be categorized to be patchy, Janus, or random, but information on length scales and geometries is not accessible.

Small-angle X-ray and neutron scattering (SAXS/SANS) have been widely used to characterize proteins and biomacromolecular complexes[17,18]. SANS is especially suitable to study multi-component systems. Thanks to the tremendous difference in the scattering cross section between hydrogen and deuterium, it is possible to highlight organic LS using contrast variation. Till now, most of the SANS on nanoparticles either simply study the density and thickness of LS[19] or provide a qualitative identification of the LS organization (Janus[20] and stripe-like[13]). On the contrary, SANS has been routinely used to identify the folding and structure of biomacromolecular complexes[21]. Typically, biomacromolecules are selectively deuterated and/or measured in solvents with varying deuteration content to obtain datasets of multiple contrasts. These curves are fitted with different approaches; among all, the program MONSA has been one of the most successful algorithms to retrieved three-dimensional (3D) structures through Monte Carlo methods[22].

In this paper, we show that this approach can be translated to derive the LS structure on nanoparticles. Through a systematic study of various monodispersed nanoparticles, we establish MONSA with contrast variation SANS as a tool to quantitatively determine complex morphologies and length scales of mixed LS. The approach is able to distinguish very similar structures. Furthermore, we demonstrate that this methodology is versatile for nanoparticles with different types of core elements as well as ligand chemistry.

## Results

**Combining SANS and Monte Carlo calculation.** In order to successfully retrieve morphological information from SANS, several conditions need to be fulfilled. First, nanoparticles must be colloidally stable in solvent, i.e., reaching ~0.1% v/v without aggregation. As a consequence, one could conveniently assume the structure factor equals to 1 during the analysis. Second, nanoparticles must be monodisperse (<10% size distribution), as high polydispersity would smear out form factor oscillations making it difficult to analyze. Third, one of the ligands in LS needs to be deuterated in order to be distinguished in multiple-component systems. Fortunately, nowadays such requirements are met for many nanoparticles[23].

To begin, we established and tested the validity of our method using highly monodisperse gold nanoparticle coated with phenylethanethiol (PET) and dodecanethiol (DDT; Fig. 1a). Based on thermodynamic considerations, these two ligands with different lengths and a strong tendency to phase separation are expected to generate patchy domains. Multiple syntheses were performed, using either both ligands hydrogenated or with one of the two ligands deuterated (hereafter we use a "d" in front of the acronym to represent a deuterated molecule). We produced PET-DDT, dPET-DDT, and PET-dDDT nanoparticles. All the three syntheses led to nanoparticles of the same size and polydispersity, i.e., $4.4 \pm 0.4$ nm in diameter, as calculated from transmission electron microscopy (TEM) and SAXS data (Supplementary Fig. 1). In addition to having the same size and size distribution, implicit in this whole methodology is that the three syntheses performed lead to the same LS composition and morphology, i.e., deuterated ligand have no effect on the nanoparticles. Furthermore, we also assume that deuteration has no effect in the solvation of the particles. If these two assumptions were not to be true, fitting, for example, the dPET-DDT and PET-dDDT scattering curves would lead either to large errors or to unphysical results. In order to test whether the whole hydrogenated nanoparticles (PET-DDT) and the selectively deuterated particles (dDDT-PET) have the same composition, we used FTIR and found no significant difference. The results are discussed and shown in Supplementary Note 1 and Supplementary Fig. 2. Furthermore, the quality of the fits throughout this work supports the fact that deuteration has minimum effect on the size and composition of nanoparticles. The LS composition was determined by $^1$H-NMR to be PET:DDT = 0.57:1 (Supplementary Fig. 1). All nanoparticles described in this work showed no sharp peaks in NMR before core-etching, i.e., almost no free ligands[24]. Therefore, the calculated ratio reflects the final ligand composition on the nanoparticles.

SANS data were acquired for dPET-DDT and PET-dDDT using deuterated toluene as solvent. Detailed experimental parameters are described in the Methods section. As shown in Fig. 1b, the scattering patterns are significantly different under the two ligand contrast conditions since different parts of LSs are probed. The evident shift of the oscillation minima from $q = 0.12$ Å$^{-1}$ in dPET-DDT to $q = 0.15$ Å$^{-1}$ in PET-dDDT means that the overall size of the nanoparticle appears much smaller when the contrast of longer ligand is matched to the solvent. The accurate chain lengths of the two ligands could be further calculated from pair distance distribution functions, $P(r)$, using GNOM package[25] (Fig. 1c). $D_{max}$ in the $P(r)$ represents the largest distance in the system, which is the overall diameter of the core-shell nanoparticles. The $D_{max}$ of PET-dDDT and dPET-DDT are 58 and 75 Å, respectively. The difference between $D_{max}$ and core

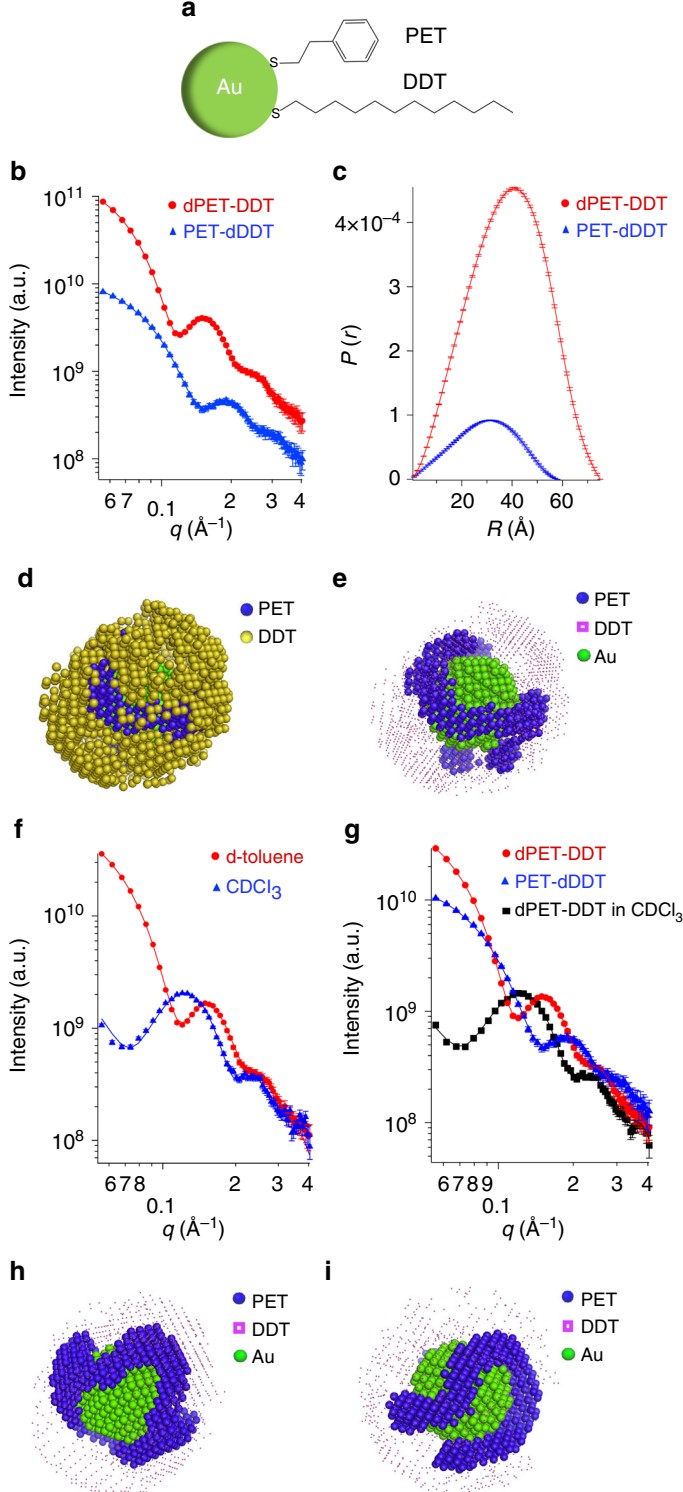

**Fig. 1** SANS data and 3D models of dPET-DDT and PET-dDDT nanoparticles. **a** Structure of PET and DDT molecules. **b** SANS data (dots) and fits (full lines) of dPET-DDT (red) and PET-dDDT (blue) nanoparticles. The fits are from MONSA fitting and correspond to the theoretical scattering curves from the model. The final $\chi^2$ value of the model fitting is 0.79, indicating the good quality of the fits. **c** Pair distance distribution function $P(r)$ of dPET-DDT (red) and PET-dDDT (blue) calculated from SANS data. **d** 3D low-resolution model built by fitting the two SANS data. All three phases, i.e., DDT (yellow), PET (blue), and gold (green), are shown. **e** Same model highlighting shorter ligands (PET) for clarity. DDT phase is marked with small pink dots. **f** SANS data and fits and from measurements with two different solvent contrasts. Data are marked in red and blue for measurements in deuterated toluene and chloroform, respectively. **g** SANS data and fits when combining three different contrasts. The resulted 3D models for the two- (**h**) and three-contrast fitting (**i**). The error bars in the SANS data represent standard deviations of neutron counts and are explained in detail in Method section

diameter comes from the LS and equals the doubled length of the ligands. Therefore, the organic LS thickness was found to be 7 and 15 Å for PET-dDDT and dPET-DDT, respectively. This difference indicates that DDT ligand is around 8 Å longer than PET in deuterated toluene. With the chain length difference of the two molecules being around six C–C bond lengths, this result indicates that DDT ligands exist in an extended conformation in toluene, which in turn explains the good colloidal stability of the nanoparticle.

After data acquisition, a multiple-phase bead model was built using MONSA. A spherical search volume (with diameter equals to $D_{max}$) was generated by close packing small beads of 2 Å radius. Between 30 and 38 Å radius only DDT and solvent beads were allowed according to the indications retrieved from $P(r)$, while beads within the central 16 Å radius were fixed to gold in order to reduce the calculation time and act as physical constraints. All other beads were free to be assigned to all four possible components. The aim of the Monte Carlo-type search implemented in MONSA was to fit simultaneously the SANS curves and minimize the overall discrepancy ($\chi^2$) between the experimental data and fits based on the theoretical spectra of the multiphase bead model. A detailed description of the algorithm and expression of $\chi^2$ is reported in the Supplementary Note 2. In order to help the readers judge the quality of the fits we have added in Supplementary Information comparisons between the experimental data and the theoretical scattering of known common ligand morphologies, Supplementary Fig. 3. Before calculation, the LS composition was extracted from NMR data and converted to a volume ratio by considering the grafting density obtained by thermogravimetric analysis (TGA; Supplementary Fig. 4). A penalty on the quality of fits was imposed when the bead ratio of the model deviates from the experimental ratio during the fitting. Nevertheless, this does not guarantee that the final ratio in the resulting model is the precisely same as the one inputted. Figure 1d, e shows the achieved model of PET-DDT nanoparticle. The PET ligand forms elongated domains from three to four beads thick. This feature is consistent across the whole nanoparticle surface, as highlighted in Supplementary Fig. 5 thanks to different orientations. Models were predicted several times starting from distinct random configurations and were all found very similar (Supplementary Fig. 6), confirming the reliability of the fitting process. Constraints on input parameters (volume fraction, core size, and looseness) were then tested (description on the fitting could be seen in Supplementary Note 3 and Supplementary Fig. 7) and all the fitting conditions led to the same morphological features, indicating the robustness of the methodology.

Moreover, we verified whether SANS data from different solvent contrasts could also be used to reconstruct nanoparticle models. The dPET-DDT sample was measured in deuterated chloroform, thus modifying the contrasts of all the components (Supplementary Table 1). Combining with the data (Fig. 1f) from deuterated toluene, a 3D model was generated (Fig. 1h) with overall morphological features and averaged thickness of domains being the same as the previous model. Furthermore, when fitting all the three datasets of different contrasts (Fig. 1g), models with consistent domain features were also accomplished (Fig. 1i), the quality of the fits remaining the same.

**Distinguishing different morphologies**. Next, we examined nanoparticles with different LS morphologies to discuss the ability of SANS to distinguish similar structures. Three mixed-ligand gold nanoparticles were measured (Fig. 2a–c), namely dDDT-DDT, 11-mercaptoundecanoic acid (MUA)-DDT, and 1-butanethiol (BT)-MUA nanoparticles. Detailed synthesis and

characterizations are reported in the SI. As illustrated in Fig. 2d–f, three distinct morphologies are determined for this set of nanoparticles. As one expects, Fig. 2d shows that DDT and its deuterated analog self-assemble in a random morphology, which represents the thermodynamic equilibrium state of this LS. For the MUA-DDT nanoparticle (Fig. 2e), with no entropic contribution from chain length mismatch and large difference in polarity, phase-separated domains (Janus morphology) were identified. In the case of the MUA-BT nanoparticle, the large contribution from chain length differences between the two different types of ligands resulted in a stripe-like structure (Fig. 2f). The SANS models were further supported by coarse-grained dissipative particle dynamics (DPD) simulations, confirming that the morphologies determined by SANS are indeed the thermodynamically favored structures, Supplementary Fig. 8.

Comparing the SANS pattern of the three nanoparticles (Fig. 2a–c), one immediately notes that the more the phases are separated the less sharp are the oscillations. This is related to the fact that LS with patchy domains has an overall lower symmetry, deviating from a concentric distribution of the components. As SANS is very sensitive to inhomogeneous distribution of scattering length densities, features of different degrees of phase separation can be easily detected. This is most emphasized in the dMUA-DDT case, where the MUA phase is matched to the solvent, and the overall shape of the nanoparticles is no longer close to spherical.

We further explored whether two rather similar morphologies could be distinguished. dPET-DDT nanoparticles were synthesized using a different approach, i.e., ligand exchange reaction. It has been demonstrated that ligand exchange reactions at room temperature often leads to nanoparticle morphologies that are not at the equilibrium[26,27]. After the ligand exchange reaction and thorough purification, the nanoparticles were annealed in pure toluene for 24 h at 50 °C to allow for the morphology to equilibrate. SANS data on nanoparticles both before and after annealing were then recorded (Fig. 2g). The scattering curves of the two nanoparticles exhibit slight differences at high angles with the oscillation feature in the annealed sample being sharper. At the same time, at low q Guinier region, the two patterns perfectly overlap. Such results indicate that the two nanoparticles have small differences in LS organization, despite the size and ligand composition are the same. The 3D model reported in Fig. 2h, i demonstrates that the nanoparticle after annealing presents slightly thinner elongated domains.

This result is particularly exciting since currently no other technique could achieve such high sensitivity. Similar capabilities have already been demonstrated in SAXS studies on proteins, where small differences in the heterogeneous or homogeneous hydration layer with only a few Å thickness could be readily captured[17]. The overall sizes of PET-DDT nanoparticles are around 7 nm, thus the organization of features with approximately 1 nm size is expected to have notable influences on the scattering length density distributions. As shown in the Supplementary Note 4 and Supplementary Fig. 9, we also quantified the distribution of the stripe thickness in each model. On average, the thickness of the PET domains changed from 1.8 ± 0.5 to 1.6 ± 0.7 nm after annealing, suggesting that upon annealing thinner stripe-like features are formed and the distribution of the domain sizes increases.

**A systematic study**. Theoretical studies have predicted the evolution of equilibrium morphologies of LS when the size and ligand ratio vary. However, currently no experimental studies have thoroughly examined such phenomena due to the limitations of the existing techniques. Here, taking advantage of the

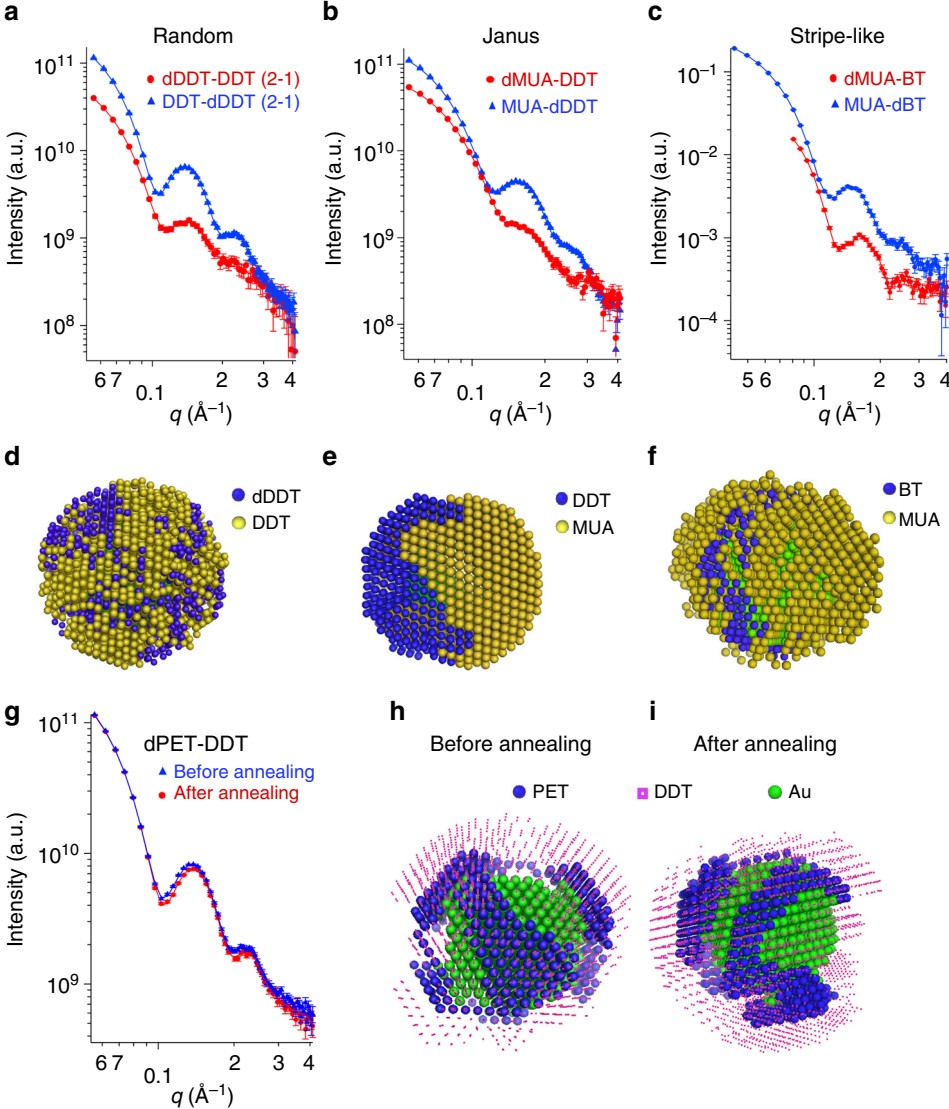

**Fig. 2** SANS data and 3D models for nanoparticles with different morphologies. SANS data and fits for dDDT-DDT (**a**), MUA-DDT (**b**), and MUA-BT (**c**) nanoparticles. For the first nanoparticle, the SANS data of dDDT-DDT are in blue and of DDT-dDDT are in red. While for the last two samples, the SANS curve is in red when MUA is deuterated and in blue when the other ligand is deuterated. The corresponding 3D models are presented in **d**–**f**. **g** SANS data and fits for dPET-DDT nanoparticle before (blue) and after (red) annealing. The corresponding models are shown in **h** and **i**, with PET beads in blue and DDT beads in pink for clarity. The error bars in the SANS data represent standard deviations of neutron counts and are explained in detail in Method section

sensitivity and quantification capability of SANS, we performed a series analysis on PET-DDT nanoparticles with various sizes and LS compositions.

The detailed synthetic procedures of the different nanoparticles are described in the SI together with basic characterizations such as NMR, TEM, SAXS, and TGA. Specifically, we measured 3.2, 4.4 (i.e. the nanoparticle described above), and 5.5 nm nanoparticles to evaluate the effect of the metal core size (Fig. 3a–c). The 3D models (Fig. 3g–i) display systematic differences among the resulting morphologies. The thickness of the domains scales inversely with the nanoparticle core sizes, as expected from computer simulation studies. The elongated domain thickness is 1.6 ± 0.3, 1.3 ± 0.3, and 1.2 ± 0.4 nm for 3.2, 4.4, and 5.5 nm nanoparticles, respectively. Then, three 4.4 nm nanoparticles were employed to explore the influence of the ligand ratio, namely PET:DDT = 0.2:1, 0.6:1, and 1.2:1 as illustrated in Fig. 3d–f, respectively. One can see from the reconstructed models (Fig. 3j–l) that the domains of PET phase evolve from patches for lower

PET ratios to stripes and finally to percolated domains for higher PET fractions. Accordingly, the dimension of the domain features evolves from 1.5 ± 0.3 and 1.3 ± 0.3 nm to 1.6 ± 0.4 nm, when the ligand ratio increases.

All models obtained from SANS were then directly compared to DPD simulation (Fig. 3m–r). The program SUPCOMB was used to superimpose the SANS reconstructions onto the simulation models in 3D[28]. The program represents each input structure as an ensemble of points, and then minimizes a normalized spatial discrepancy to find the best alignment of two structures. The alignment reported in Fig. 3s–x was achieved focusing on the PET phase. The morphologies (size and shape of the domains) from two methods present striking similarity for all the nanoparticles. As a further confirmation, we compared the thickness of the PET domains from the DPD models to that of the SANS models. As illustrated in Supplementary Fig. 9, the two statistics show good agreements.

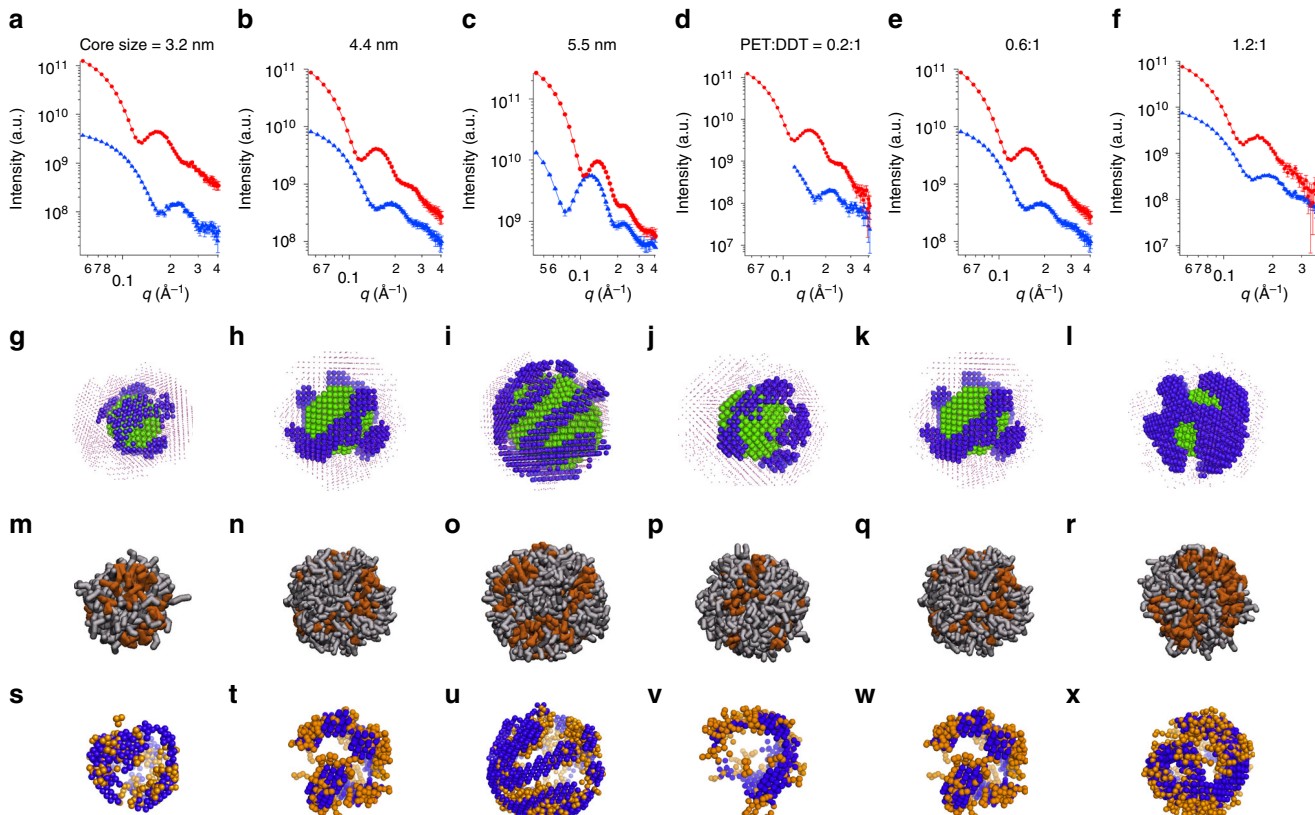

**Fig. 3** SANS and simulation study of various PET-DDT nanoparticles . In SANS data, red dots and curves represent the data and fits of dPET-DDT particles, while blue ones represent that of PET-dDDT particles. All the solvents are deuterated toluene. The only exception is in the 5.5 nm nanoparticles where the blue curve represents dPET-DDT in deuterated chloroform. The fits are from MONSA fitting and correspond to the theoretical scattering curves of each model. In simulation models, gray sticks represent DDT ligands, while orange sticks represent PET ligands. Panels **a–c** compare same ligand ratio but different core-sized AuNPs: **a** SANS data and fits (with a final $\chi^2$ value of the model fitting of 1.52), **g** model, and **m** mesoscale morphology of 3.2 nm particle. **b** SANS data and fits (with a final $\chi^2$ value of the model fitting of 0.79), **h** model, and **n** mesoscale morphology of 4.4 nm particle. **c** SANS data and fits (with a final $\chi^2$ value of the model fitting of 1.01), **i** model, and **o** mesoscale morphology of 5.5 nm particle. Panels **d–f** compare AuNPs with same 4.4 nm core size but with different ligand ratios: **d** SANS data and fits (with a final $\chi^2$ value of the model fitting of 0.94), **j** model, and **p** mesoscale morphology of PET:DDT = 0.2:1 particle. **e** SANS data and fits (with a final $\chi^2$ value of the model fitting of 0.71), **k** model, and **q** mesoscale morphology of PET:DDT = 0.6:1 particle. **f** SANS data and fits (with a final $\chi^2$ value of the model fitting of 0.71), **l** model, and **r** mesoscale morphology of PET:DDT = 1.2:1 particle. 3D overlapping comparisons of the PET phase of SANS and simulation model, generated using SUPCOMB program are in **s-x**. The error bars in the SANS data represent standard deviations of neutron counts and are explained in detail in Method section

**Polydispersity and ensemble-based analysis**. The method developed here is powerful. It allows for the determination of the LS morphology with a precision that no other technique can reach. One limitation is the need for at least one type of ligand to be deuterated . While this will add a cost to the method, as for the synthetic challenges, there is a growing number of deuteration facilities in the world that can provide the needed molecules. The main limitation we envisage is the requirement on high mono-dispersity. Unlike proteins, nanoparticles usually have a certain polydispersity, which involves the core size, LS composition, and LS morphology. We will address the influence of these three polydispersity on SANS interpretation in the following paragraphs.

Currently, many techniques are available to accurately determine the distribution of nanoparticle core sizes. It has been found that 10% polydispersity in nanoparticle dimension has only little effects on the quantitative shape retrieval from SAXS data[29]. Both TEM and SAXS confirmed <10% core size distribution for all our samples. Instead, to date, no method is available to assess the polydispersity of the LS composition among nanoparticles. The only studies on the argument come from property-based studies and mass spectrometry results on nanoclusters[30–32]. In

both cases, LS composition and organization variations were found to be narrow. In the presence of a distribution of either size or LS composition, since the resulting scattering pattern is averaged over all the form factors, the oscillation features of SANS data become smeared. Such effect resembles the result of having larger phase-separated domains, which gives a less symmetric scattering profile. Therefore, the model analysis of SANS data would lead to an over-estimation of the length-scale of phase separation.

Second, we would like to address the issue regarding the polydispersity on LS morphologies. We applied an ensemble-based analysis to try to fit SANS data with a library of nanoparticles of various morphologies using the OLIGOMER package[33]. Ensemble-based analysis is a powerful technique in interpreting averaged scattering data, e.g., in studying conformations of proteins in solution. In Fig. 4a are illustrated the results obtained after fitting the theoretical pattern of a perfect stripe-like nanoparticle using a library of possible morphologies. The ensemble fitting finds contributions from the two morphologies that are most similar to the theoretical one. Neither Janus nor random morphology was found to be present in the fitting. In addition, the quality of the fitting is not as good as the MONSA

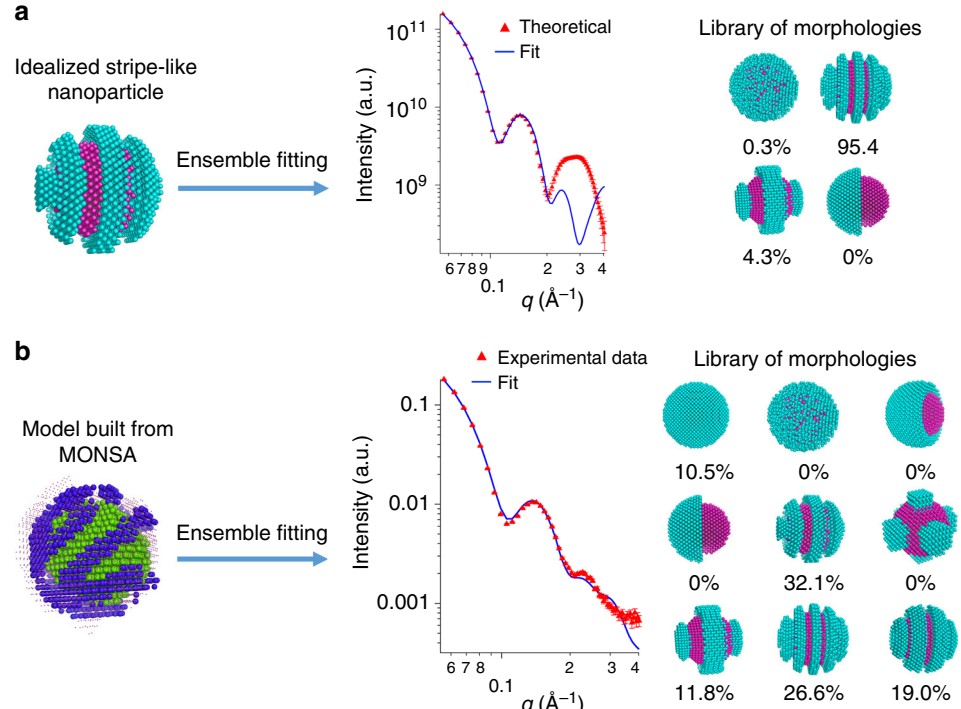

**Fig. 4** Fitting of the SANS data with a library of morphologies. The SANS pattern of idealized stripe-like structure (**a**), and 5.5 nm dPET-DDT nanoparticles (**b**) are fitted using the OLIGOMER package. The experimental data are represented by the red curve while the fitting result is the blue curve. The contribution from each morphology in the library is listed. The $\chi^2$ values for the fittings are 112.6 for the idealized stripe-like structure and 19.4 for dPET-DDT nanoparticle

fitting at high scattering angles, which confirms again that the scattering pattern of a specific morphology does not simply equal the combination of other types of morphologies.

We then applied this ensemble method to analyze our experimental data. In Fig. 4b, a variety of nanoparticle models with different morphologies were used as a library to fit the experimental PET-DDT data. The library includes homoligand, random mixed, Janus, and stripe-like nanoparticles with several stripe thicknesses. The result of the ensemble fitting highlights that most of the contributions (89.5%) is coming from similar stripe-like nanoparticles with different thicknesses. Indeed, 32.1% and 26.6% contributions come from two stripe-like nanoparticles that share similar stripe thickness with that retrieved from the MONSA fitting. The program does not recognize any contribution from Janus and random type morphologies. This ensemble analysis yields overall good quality fitting, but small differences could still be found at high scattering angles.

Overall, the results demonstrate the accuracy of the proposed approach. The evidence that only the nanoparticles with features similar to those present in the MONSA model contribute to the ensemble fitting confirms that the multiphase reconstructions are representative of the whole sample.

**Other metal nanoparticles and ternary LS**. In order to demonstrate the versatility of this method, we then analyzed two samples of nanoparticles with other types of core material. First, we focused on copper nanoparticles coated with a mixture of tetradecylphosphonic acid and dDDT. Detailed synthetic procedures and characterization can be found in the Supplementary Note 5. Since copper nanoparticles are quite sensitive to oxidization, they could not be easily characterized by techniques such as STM and mass spectrometry to access their LS morphology. Thanks to the versatile sample conditions achievable in SANS measurements, we successfully measured the scattering data of

these nanoparticles (Fig. 5a). Interestingly, the 3D model shows a patchy-type organization similar to that characterized for gold nanoparticles (Fig. 5d).

Next, we performed the same measurements on silver nanoparticles. It has been shown previously that silver nanoparticles can be properly characterized by MALDI mass spectrometry[34]. Information from various silver-thiolate fragments acted as an indicator of the degree of phase separation. We synthesized and measured dPET- and DDT-coated silver nanoparticles as described in the Supplementary Note 5. Figure 5b, e show the corresponding scattering profiles and the 3D reconstruction, in which a mixture of patchy and thin domain features can be appreciated. Theoretical fragmentation patterns for MALDI spectra were then calculated starting from this bead model. As demonstrated in the Supplementary Fig. 10, the calculated fragmentation matches well with the experimental values, with only small discrepancies, hence supporting the reliability of the predicted multiphase model.

Finally, we applied SANS to study more complex surface structures, namely nanoparticles protected with a mixture of three different ligands. The morphologies of ternary LS nanoparticles have been predicted theoretically previously[35]. However, none of the current characterization techniques could be used to determine those structures. Here we characterize, for the first time, the morphology of a ternary mixed-ligand gold nanoparticle with a 4.4 nm size core. The SANS data were collected in deuterated toluene for all PET-dOT-DDT, PET-OT-dDDT, and dPET-OT-DDT systems (Fig. 5c). A four-phase 3D model was then calculated using MONSA simultaneously fitting the three SANS curves. The obtained reconstruction exhibits interconnected patchy structures (Fig. 5f), with PET forming some small islands, while OT and DDT phases being stripe-like. Molecular calculations were then employed to validate such findings. As illustrated in Supplementary Fig. 11, computer

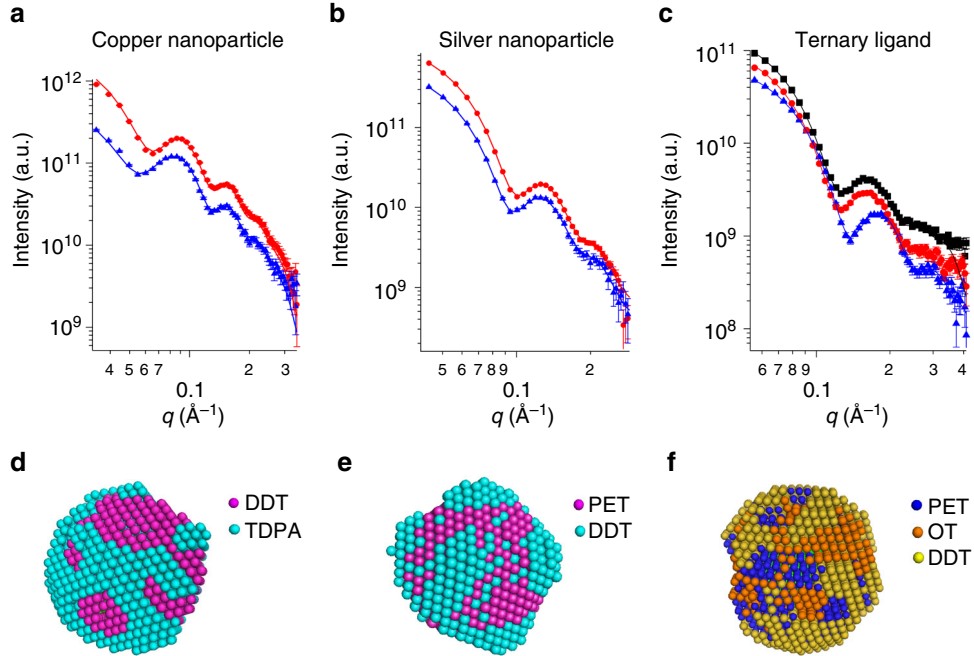

**Fig. 5** SANS data and models for different types of nanoparticles. . **a** SANS data and fits for TDPA- and dDDT-coated copper nanoparticles. The red curve stands for the measurements in deuterated toluene while blue curve is for the measurements in a mixture of deuterated (80%) and hydrogenated (20%) toluene. Corresponding 3D model is in **d** with the blue beads stand for longer ligand (TDPA) and red stands for shorter ligand (dDDT). **b** SANS data and fits for dPET- and DDT-coated silver nanoparticles. The solvent conditions are the same as copper nanoparticle samples. Corresponding 3D model is in **e** with the blue beads stand for longer ligand (DDT) and red stands for shorter ligand (dPET). **c** SANS data and fits for ternary LS gold nanoparticles, in which red dots and curves represent the data and fits of PET-dOT-DDT nanoparticles, blue ones are for PET-OT-dDDT nanoparticles, and black ones are for dPET-OT-DDT nanoparticles. The solvents for all the measurements are deuterated toluene. Corresponding 3D model is in **f**. Yellow, orange, and blue beads stand for DDT, OT, and PET ligands, respectively. The error bars in the SANS data represent standard deviations of neutron counts and are explained in detail in Method section

simulations confirm that the shortest ligand typically forms isolated regions; elongated domains of OT are also easily discernible surrounded by DDT chains.

We demonstrated here that SANS combined with 3D reconstruction is a powerful and versatile technique for the characterization of LS morphologies on nanoparticles. With the unique opportunity to use contrast variation, SANS is capable of studying complex nanostructures and is applicable to different types of nanoparticles. Morphologies varying from random, Janus to complicated patchy/stripe-like structures can all be distinguished quantitatively with high sensitivity. We believe that this technique could become a general tool in nanoparticle research.

## Methods

**General**. Deuterated DDT and OT were purchased from CDN Isotopes, Inc. All other chemicals were purchased from Sigma-Aldrich and used as received. TEM images were taken using Philips CM12 operating at 100 kV. NMR spectra were recorded using Bruker 400 MHz spectrometer using CDCl₃ as solvents. SAXS measurements were done using Rigaku BioSAXS 2000 machine. TGA measurements were done using TGA 4000 instrument from Perkin Elmer. Around 15 mg of nanoparticle samples were used for each measurement. The flow of nitrogen gas is at 20 ml/min and the samples were heated from 50 to 750 °C at 5.0 °C/min. The MALDI analysis on silver nanoparticle was performed using the Bruker AutoFlex Speed instrument. Chloroform solutions of around 5 mg/ml nanoparticles were prepared and mixed with an equal volume of DCTB matrix solution (20 mg/ml in chloroform). For each sample, 2 μl aliquot of such solution mixture was deposited and dried onto a stainless ground steel target plate. Measurements were performed in positive ionization mode and operated in the linear mode in the 700–3500 $m/z$ mass range. The laser intensity was kept at around 30% for all measurements. Mass spectra were processed with FlexAnalysis (Bruker) software.

**Synthesis and characterization of gold nanoparticles**. The nanoparticle synthesis followed the protocol reported by Stucky et al. with solvents and feed ratios of ligand mixtures varied. For 5.5 nm nanoparticles, toluene was used as solvent; for 3.2 nm nanoparticles, chloroform was used as solvent; and for 4.4 nm

nanoparticles, a 1:1 mixture of toluene and chloroform was used. The ligand ratio on nanoparticle surfaces is controlled by the feed ratio during the reaction. In general, 123 mg triphenylphosphinegold (I) chloride and 0.25 mmol ligand mixtures were dissolved in the 40 ml corresponding solvents and heated to 70 °C. Then, 217 mg borane t-butylamine complex was added into the solution under rapid stirring. A volume of 40 ml methanol was added to quench the reaction after 1 h. Nanoparticles were precipitated overnight and then purified by repeated centrifugation and washing with acetone. The final black precipitates were dried in vacuum overnight.

The ligand density on nanoparticle surfaces were measured by TGA, Supplementary Fig. 5. The size and polydispersity of the nanoparticles were characterized by TEM and SAXS as shown in Supplementary Information. TEM analysis is based on statistics of more than 500 nanoparticles. ¹H-NMR of the nanoparticles, Supplementary Fig. 3, before and after etching the cores were checked to make sure the cleanness (free from unbounded ligands) and final ligand ratio on nanoparticle surfaces.

**SANS measurements**. SANS measurements of PET-DDT, DDT homoligand, and ternary ligand-protected gold nanoparticles are mainly conducted on the SANS-I instrument at Paul Scherrer Institute, while the SANS data of the 5.5 nm PET-DDT nanoparticle, the MUA-BT and MUA-DDT nanoparticles, and copper and silver nanoparticles were recorded on KWS-2 at Jülich Center for Neutron Science. Measurements were performed at 20 °C, using 1.5 m sample-to-detector distance, at 5 Å wavelength with a collimation setup of 6 m and a $q$ range from 0.05 to 0.4 Å⁻¹. The sample concentration was approximately 10 mg/ml corresponding to a volume fraction of gold nanoparticles solution of around 0.1%. The two-dimensional scattering data were processed and reduced using BerSANS software, including radial averaging, background subtraction, empty cell and transmission correction, and normalization to an absolute scale. The error bars, i.e., uncertainties of the intensity, represent the standard deviation of the counts in the pixels within each bin. The propagation of the errors from solvent subtraction is given by $\sigma_{sub} + \sqrt{\sigma_{sample}^2 + \sigma_{solvent}^2}$.

**DPD calculations**. Coarse-grained methods as DPD (see Supplementary Note 6 for details on DPD methodology) are especially suitable for predicting polymer phase separations and self-assembling phenomena as those involved in the organization of mixtures of ligands on solid curved surfaces, since they allow sampling much longer times with respect to atomic resolution techniques (e.g., molecular

dynamics) while retaining a molecular description of the self-assembled monolayer. The computational procedure employed here is based on a predictive multiscale molecular simulation protocol, i.e., a combination of atomistic/coarse-grained calculations, and was already successfully applied by us to investigate the self-assembling organization of several immiscible ligand mixtures, including poly (ethylene oxide) terminating hydrocarbon/perfluorocarbon thiolated chains as well as mercaptoundecanesulfonate/octanethiol ligands, on spherical gold nanoparticle surfaces[36]. Full details of molecular models and simulations are given in the SI.

Briefly, the initial structure of the nanoparticle core was constructed by arranging gold coarse-grained units (or DPD beads) on an fcc lattice into an icosahedron of the desired diameter. Each ligand was represented by a flexible chain model of beads connected by harmonic springs whose topology was assessed by matching the atomistic and mesoscale pair correlation functions for each ligand chain. Thiols were placed close to the nanoparticle surface and oriented outward with the head-tail vector along the radial direction using the Packmol package[37]. The appropriate number of ligands was inserted in order to reproduce the experimental number of molecules for $nm^2$ of surface. A random configuration was imposed to arrange the chains on the gold surface. Then, the modified NP was solvated. Each system was tested on three independently generated starting configurations. All gold beads were forced to move as a rigid body during the calculation, while the sulfur heads could diffuse laterally on the NP surface. To ascertain that the morphologies obtained correspond to a thermodynamic equilibrium, simulations were performed starting also from a phase-separated configuration of the ligands.

Each initial configuration was first relaxed for $1 \times 10^4$ steps and a time step of $\Delta t = 0.01\tau$. Then, at least additional $6 \times 10^6$ time steps ($\Delta t = 0.02\tau$) were performed for productive runs. System equilibration was assessed monitoring temperature, pressure, density, and potential energy behavior as well as composition of nearest neighbors of head groups.

All mesoscale production runs, analysis, and imaging were performed using the software Large-scale Atomic/Molecular Massively Parallel Simulator (LAMMPS) running on graphics processing units and the software Visual Molecular Dynamics.

**Synthesis of deuterated PET and MUA**. The deuterated PET and MUA ligands are provided by National Deuteration Facility of ANSTO. Detailed synthetic procedures as well as characterization could be found in Supplementary Note 7 and Supplementary Fig. 12-37.

**Data availability**. All data are available from the authors upon reasonable request. The raw data of all the SANS experiments described here can be found through the following link: https://figshare.com/projects/Raw_data_for_the_characterization_of_mixed_ligand_nanoparticles/29398

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

## Acknowledgements

Z.L., Q.K.O., and F.S. gratefully acknowledge funding from the Swiss National Foundation Division II. D.M. and P.P. are grateful to the Italian Ministry of University Research (MIUR) through the Scientific Independence of Young Researchers (SIR) project "Structure and function at the nanoparticle biointerface" (Grant RBSI14PBC6) for its generous financial support. The National Deuteration Facility is partly supported by the National Collaborative Research Infrastructure Strategy – an initiative of the Australian Government. The authors are grateful to PSI and JCNS for kindly providing valuable beam times. We thank the helpful discussion with Dr. Mauro Moglianetti and Dr. Benjamin Le Ouay. We thank the help from Jian Wang for helping with TGA measurements and Yue Wang for MALDI measurements.

## Author contributions

Z.L. synthesized and characterized gold and silver nanoparticles; A.L. and R.B. synthesized copper nanoparticles; Z.L., J.K., and A.R. performed SANS experimental studies; Z. L. and D.I.S. performed ab initio fitting on SANS data; Q.K.O. performed STM studies; D.M. and P.P. performed molecular dynamic studies; Z.L. and S.B. performed SAXS analysis; A.K.-H., Z.L., and T.D. synthesized deuterated ligands; Z.L., D.I.S. and F.S. designed research; Z.L., D.I.S., P.P., F.S. prepared the manuscript.

## Additional information

**Competing interests:** The authors declare no competing interests.

