## [Peer Review File · Nature Communications]

Reviewers' comments:

Reviewer #1 (Remarks to the Author):

This article presents a method using small angle neutron scattering and selectively deuterating ligands and solvents to obtain information on the arrangement of ligand monolayers on the surface of nanoparticles. Although this is not complete structural information, it is still a valuable aid in our understanding of these particles and for how they work in the many research fields that use them. I know of no other way this or similar information can be obtained for the class of materials studied.

The article is well written and presents appropriate details for the many steps used to obtain this information. Since self-assembled monolayer protected nanoparticles are so ubiquitously used, I think this article is appropriate for publication in Nature Communications and recommend that it be published.

Mark Sutton
Physics, McGill University

Reviewer #2 (Remarks to the Author):

The manuscript from Z. Luo et al. deeply investigate the use of SANS to study the organization of ligands in mixed monolayer nanoparticles. This is a crucial subject since at the moment no general technique is available to characterize the monolayer morphology while potential applications of mixed monolayer nanoparticles are huge. The use of the SANS technique was already proposed by some of the authors in 2014 (ref. 15) and has been here extended to the investigation of mixed monolayer coated nanoparticles of different size, different monolayer composition, different core material and even ternary coatings. The issue of size and monolayer composition polydispersity is considered and analyzed. Patterns obtained by fitting of the SANS data were found to be in agreement with the results of computational simulations and in the case of silver nanoparticles, with MALDI experiments. On my opinion this is a very important paper that deserves publication on Nat. Commun. after minor revision.

1) The possibility that this method may become a general tool in nanoparticle research is claimed by the authors. However, while the technique is really powerful and versatile, general use may be heavily hampered by the need of deuterated thiols. Isotopic labelling is nowadays quite easy in protein studies, thanks to the different protein expression methods available. In the case of organic molecules, and in particular of customized nanoparticle ligands, cost and synthetic accessibility can be a major problem. This should be discussed.

2) Some assumptions implicit in the manuscript should be made explicit and if necessary discussed. In particular it is assumed that composition of nanoparticles containing a deuterated ligand is the same as the corresponding non deuterated one. This is likely true, but the use of this assumption should be stated or verified (ligand ratio in the case of deuterated thiols could be investigated by quantitative ^{13}C NMR after ligand stripping)

3) χ values ranging from 0.7 to 1.5 are reported in the paper. However, beside the point that smaller is better, it is not easy to assign a fit quality meaning to such figures. To increase the clarity of the presentation, and better guide readers not expert in SANS experiments, the results of fitting (including the obtained χ values) the SANS data with pre-determined ligand distribution models (i.e. janus, random, striped, patchy) might be shown, at least in the SI. A similar approach was used by the authors in ref. 15. In that case however, only a janus and random distributions were used and resulting χ values were not reported.

4) The previous issue is partly addressed in this manuscript when the problem of morphology polydispersion is considered and addressed by evaluating the potential contribution of combinations of different morphologies. Also in this case (bad fitting) χ values should be reported. In addition, to the several conformation considered (random, janus, striped) at least a patchy, or small islands, one should be added.

5) Again to increase the clarity of presentation, a reorganization of the SI should be considered by grouping all the nanoparticles characterising data, followed by SANS experiments and their fittings. As an instance, 6 TEM pictures are reported in figure S1 (over 10 different nanoparticles investigated in the paper), 5 SAXS and TEM size distribution are reported in figure S2 (and in different order), 3 NMR in figure S3 and 3 TEM in figure S5 (again with different orders). It is quite difficult in this way to link the pictures and compare the particles. Presentation of a full set of characterization data for each NP would be in my opinion more appropriate and easy to read.

6) The authors explain the different patterns found in term of entropic (entropy of mixing and conformational entropy) and enthalpy (inter ligand interactions) considerations. This is supported by previous investigations and the computational simulation reported here. However, it is surprising to note how the often preferred configuration is a kind of large equatorial strip separating two large patches of the other thiol. The gain of this configuration with respect to the janus one is not self-evident, since there is only a relatively small increase of boundary molecules between the two phases.

Minor points:

1) Numbering of SI figures: in the text is cited an unexciting Figure S23, while in SI Figure S8 is cited as S7.

2) Assignment and solvent of NMR spectra in figure S3 should be indicated. There is a second signal in the I2 stripped experiments at 0.85 ppm that may affect the composition determinations. Integrals of the signal at 3-2.5 ppm are apparently smaller than one may expect (about 4 instead the about 6).

3) Experimental details of TGA and MALDI experiments are not given. In the experimental section is reported a STM instrument not used and not the MALDI used.

Fabrizio Mancin
Università di Padova

Reviewer #3 (Remarks to the Author):

The authors present a new approach to directly measure the nanostructure of a multicomponent ligand shell on a solid nanoparticle core using a series of SANS measurements that are fit with Monte Carlo based calculations. This work appears to be very thorough and well done. It will be of great interest both to the scattering community, which may want to extend this approach toward fitting other complicated multicomponent systems, and the colloidal science community, for which there is currently no existing method to determine complex ligand layer structures with reasonable certainty. I recommend that this paper be published with only minor revisions related to clarity of the figures.

It would be helpful if legends could be included in all the plots where possible. The figures with legends are easier to interpret, as the same red and blue circles and triangles are used throughout the manuscript. In Figure 1 the labeling of the sub-figures is not intuitive. For the SANS data, the error bars should be defined.

Overall, I found this paper to be exceptional and very interesting. I think it could possibly even initiate a shift in the way the SANS community analyzes and interprets data for complicated systems.

Response to the Reviewers' comments

Reviewer #1 (Remarks to the Author):

This article presents a method using small angle neutron scattering and selectively deuterating ligands and solvents to obtain information on the arrangement of ligand monolayers on the surface of nanoparticles. Although this is not complete structural information, it is still a valuable aid in our understanding of these particles and for how they work in the many research fields that use them. I know of no other way this or similar information can be obtained for the class of materials studied.

The article is well written and presents appropriate details for the many steps used to obtain this information. Since self-assembled monolayer protected nanoparticles are so ubiquitously used, I think this article is appropriate for publication in Nature Communications and recommend that it be published.

Mark Sutton
Physics, McGill University

We thank the referee for his positive assessments on the importance of our work.

Reviewer #2 (Remarks to the Author):

The manuscript from Z. Luo et al. deeply investigate the use of SANS to study the organization of ligands in mixed monolayer nanoparticles. This is a crucial subject since at the moment no general technique is available to characterize the monolayer morphology while potential applications of mixed-monolayer nanoparticle are huge. The use of the SANS technique was already proposed by some of the authors in 2014 (ref. 15) and has been here extended to the investigation of mixed monolayer coated nanoparticles of different size, different monolayer composition, different core material and even ternary coatings. The issue of size and monolayer composition polydispersity is considered and analyzed. Patterns obtained by fitting of the SANS data were found to be in agreement with the results of computational simulations and in the case of silver nanoparticles, with MALDI experiments. On my opinion this is a very important paper that deserves publication on Nat. Commun. after minor revision.

We thank the referee for the comprehensive summary and positive evaluation of our work.

1) The possibility that this method may become a general tool in nanoparticle research is claimed by the authors. However, while the technique is really powerful and versatile, general use may be heavily hampered by the need of deuterated thiols. Isotopic labelling is nowadays quite easy in protein studies, thanks to the different protein expression methods available. In the case of organic molecules, and in particular of customized nanoparticle ligands, cost and synthetic accessibility can be a major problem. This should be discussed.

We agree with the referee that deuterated molecules are often not commercially available and/or expensive. However, many neutron facilities have established chemical deuteration services that are available for users. In particular, this year witness the initiation of European Deuteration Network (<https://deuteration.net>), which is funded by EU and aims at providing complex deuterated small molecules for neutron scattering users.

The following text has been added to the manuscript on page 8 to clarify this point:

"One limitation is the need for at least one deuterated ligand molecule. While this will add a cost to the method, as for the synthetic challenges, there are a growing number of deuteration facilities in the world that can provide the needed molecules."

2) Some assumptions implicit in the manuscript should be made explicit and if necessary discussed. In particular it is assumed that composition of nanoparticles containing a deuterated ligand is the same as the corresponding non deuterated one. This is likely true, but the use of this assumption should be stated or verified (ligand ratio in the case of deuterated thiols could be investigated by quantitative ^{13}C NMR after ligand stripping)

The assumption we made on the composition of deuterated molecules has been also reported and verified previously by MALDI (Harkness et. al. Angew. Chem. 2011). We found that performing quantitative \$^{13}\text{C}\$ NMR was not trivial, and consequently preferred using FTIR. As described and shown in supplementary materials, the FTIR spectra of a mixture of DDT and dDDT homoligand nanoparticles were recorded at varying molar ratios of the nanoparticles. The ratio of the intensity of the \$\text{CH}_2\$ and \$\text{CD}_2\$ stretching peaks was then calculated and plotted against the molar ratio of the nanoparticles in order to build a calibration curve. Then the FTIR

spectra of PET-DDT and PET-dDDT nanoparticles were recorded. The CD₂ peak intensity of the PET-dDDT nanoparticles was then converted to the corresponding CH₂ intensity using the calibration curve discussed above and shown in Figure S2. The ratio of between the intensity of the aromatic CH stretching and the aliphatic CH₂ stretching was then calculated for both the PET-DDT and PET-dDDT nanoparticles. For the former it was found to be 0.14 and for the latter it was found to be 0.13. We believe that these two values are within error one to each other indicating that the two particles have the same composition. We notice that these ratios do not indicated the stoichiometry on the ligand shell as they have not been corrected for the relative intensity of the two types of the peaks.

We have also added the in the main manuscript the following text to render our hypothesis explicit.

“In addition to having the same size and size distribution, implicit in this whole methodology is that the three syntheses performed lead to the same ligand shell composition and morphology, *i.e.* deuterated ligand have no effect on the nanoparticles. Furthermore, we also assume that deuteration has no effect in the solvation of the particles. If these two assumptions, when comparing for example the dPET-DDT and PET-dDDT nanoparticles, were not to be true, then fitting would not be possible in our models or lead to unphysical results. In order to test whether the whole hydrogenated nanoparticles (PET-DDT) and the selectively deuterated particles (dDDT-PET) have the same composition, we used FTIR and found no significant difference. The results are discussed and shown in Supplemental Material (see Figure S2).”

3) χ values ranging from 0.7 to 1.5 are reported in the paper. However, beside the point that smaller is better, is not easy to assign a fit quality meaning to such figures. To increase the clarity of the presentation, and better guide readers not expert in SANS experiments, the results of fitting (included the obtained χ values) the SANS data with pre-determined ligand distribution models (*i.e.* janus, random, striped, patchy) might be shown, at least in the SI. A similar approach was used by the authors in ref. 15. In that case however, only a janus and random distributions were used and resulting χ values were not reported.

$$\chi^2 = \frac{1}{n-1} \sum_{k=1}^n \left[\frac{I_{\text{exp}}(q_k) - I_{\text{calc}}(q_k)}{\sigma(I_{\text{exp}}(q_k))} \right]^2$$

The χ value we reported is actually χ^2 and is defined as:

It is used to evaluate the the statistical similarity between experimentally obtained intensities, $I_{\text{exp}}^{(k)}(q)$, and those computed from a model, $I_{\text{calc}}^{(k)}(q)$. Therefore, the smaller the χ^2 , the better fit is. However, given that the experimental intensity is limited by the error in the measurement, the χ^2 value also depends on experimental errors, $\sigma^{(k)}(q)$. If errors are correctly estimated, then the $I_{\text{exp}}^{(k)}(q)$ would be, on average, one $\sigma^{(k)}(q)$ from the $I_{\text{calc}}^{(k)}(q)$. Hence the ideal χ^2 should be close to 1. Otherwise, if it is too small, there is the risk of over-fitting. In the case of neutron scattering experiments, the true values of errors of intensity is hard to estimate. In fact, the errors are very often over-estimated due to reasons such as the low intensity of neutron beams. Therefore, we did not focus on the χ^2 values in the manuscript. But all of the fit presented show reasonably good χ^2 value, *i.e.* not too much deviated from 1, and should be judged by the reader in the graphical form provided.

However, we agree with the referee that showing the χ^2 value of pre-determined models could act as a good reference. In the Figure S3, we added the comparison between the experimental data against pre-determined models (Janus, random, stripe-like and patchy) together with the χ^2 values.

We have added the following to the manuscript:

“In order to help the readers judge the quality of the fits we have added in supplementary materials comparisons between the experimental data and the theoretical scattering of known common ligand morphologies, Figure S3.”

4) The previous issue is partly addressed in this manuscript when the problem of morphology polydispersion is considered and addressed by evaluating the potential contribution of combinations of different morphologies. Also in this case (bad fitting) χ values should be reported. In addition, to the several conformations considered (random, janus, striped) at least a patchy, or small islands, one should be added.

We have added in the library of the ensemble calculations one patchy nanoparticle. The new fitting is updated in Figure 4B. The χ^2 values are also reported.

5) Again to increase the clarity of presentation, a reorganization of the SI should be considered by grouping all

the nanoparticles characterising data, followed by SANS experiments and their fittings. As an instance, 6 TEM pictures are reported in figure S1 (over 10 different nanoparticles investigated in the paper), 5 SAXS and TEM size distribution are reported in figure S2 (and in different order), 3 NMR in figure S3 and 3 TEM in figure S5 (again with different orders). It is quite difficult in this way to link the pictures and compare the particles. Presentation of a full set of characterization data for each NP would be in my opinion more appropriate and easy to read.

We have re-organized the presentation of the characterization data in the SI according to the referee's suggestion. NMR data are shown together with SAXS and TEM data. The characterizations on each AuNP are shown in full set.

6) The authors explain the different patterns found in term of entropic (entropy of mixing and conformational entropy) and enthalpy (inter ligand interactions) considerations. This is supported by previous investigations and the computational simulation reported here. However, it is surprising to note how the often preferred configuration is a kind of large equatorial strip separating two large patches of the other thiol. The gain of this configuration with respect to the janus one is not self-evident, since there is only a relatively small increase of boundary molecules between the two phases.

There are two answers to this question. The first is that the referee may be confused by our presentation as the single large equatorial 'stripe' is presented on 2 out of the 11 nanoparticles we studied (but they are in fact shown more than once). Hence one cannot define this a high occurrence. Second, for both cases the simulation presented in the paper show the same arrangement. We believe that this arrangement is explained in terms of balance between entropy determined at the interface and enthalpy of phase separation, but this paper is not about this, hence we prefer not to speculate further.

Minor points:

1) Numbering of SI figures: in the text is cited an unexciting Figure S23, while in SI Figure S8 is cited as S7.

We have corrected these numbering problems.

2) Assignment and solvent of NMR spectra in figure S3 should be indicated. There is a second signal in the I2 stripped experiments at 0.85 ppm that may affect the composition determinations. Integrals of the signal at 3-2.5 ppm are apparently smaller than one may expect (about 4 instead the about 6).

The integration of the relevant peaks as well as the assignments are plotted in the new version of supplementary information, Figure S1. For the calibration, we used the largest peaks, i.e. the 9 CH₂ in the alkane-chain of DDT instead of the methyl group at around 0.88 ppm. On the other hand, the integration for the methyl-group is always around 3. The integrals at 3-2.5 ppm are the sum of the two CH₂ peaks of the PET and the CH₂ near disulfide in DDT. The reason why it is around 4 instead of 6 is that the ratio of PET to DDT is around 0.5: 1.

3) Experimental details of TGA and MALDI experiments are not given. In the experimental section is reported a STM instrument not used and not the MALDI used.

The details about the TGA and MALDI experiments are provided in Materials and Method section and the STM description is removed. We have added the following text:

"TGA measurements were done using TGA 4000 instrument from Perkin Elmer. Around 15 mg of nanoparticle samples were used for each measurement. The flow of nitrogen gas is at 20 ml/min and the samples were heated from 50 °C to 750 °C at 5.0 C/min. The MALDI analysis on silver nanoparticle was performed using the Bruker AutoFlex Speed instrument. Chloroform solutions of around 5mg/ml nanoparticles were prepared and mixed with an equal volume of DCTB matrix solution (20 mg/ml in chloroform). For each sample, 2 µl aliquot of such solution mixture was deposited and dried onto a stainless ground steel target plate. Measurements were performed in positive ionization mode and operated in the linear mode in the 700-3500 m/z mass range. The laser intensity was kept at around 30% for all measurements. Mass spectra were processed with FlexAnalysis (Bruker) software."

Reviewer #3 (Remarks to the Author):

The authors present a new approach to directly measure the nanostructure of a multicomponent ligand shell on a solid nanoparticle core using a series of SANS measurements that are fit with Monte Carlo based

calculations. This work appears to be very thorough and well done. It will be of great interest both to the scattering community, which may want to extend this approach toward fitting other complicated multicomponent systems, and the colloidal science community, for which there is currently no existing method to determine complex ligand layer structures with reasonable certainty. I recommend that this paper be published with only minor revisions related to clarity of the figures.

We thank the referee for giving credit to the importance of the work for both nanoparticle and scattering community.

It would be helpful if legends could be included in all the plots where possible. The figures with legends are easier to interpret, as the same red and blue circles and triangles are used throughout the manuscript. In Figure 1 the labeling of the sub-figures is not intuitive.

We have added applicable legends in all of the figures of the manuscript. The labelling in Figure 1 has also been modified.

For the SANS data, the error bars should be defined.

We have added the following sentence in the main text:

“The error bars, i.e. uncertainties of the intensity, represent the standard deviation of the counts in the pixels within each bin. The propagation of the errors from solvent subtraction is given by: $\sigma_{sub} = \sqrt{\sigma_{sample}^2 + \sigma_{solvent}^2}$.”

Overall, I found this paper to be exceptional and very interesting. I think it could possibly even initiate a shift in the way the SANS community analyzes and interprets data for complicated systems.

Reviewers' Comments:

Reviewer #2:

Remarks to the Author:

I've already commented on the high relevance of the manuscript. On my opinion, the revised manuscript addresses convincingly all the points raised by the referees and can be published as it is.

In particular, the FT-IR investigation performed on deuterated and non-deuterated mixed-monolayer nanoparticles supports the expectation that they have the same composition. The supporting information is much clearer.